

# An exploration of the effect of proprioceptive knee bracing on biomechanics during a badminton lunge to the net, and the implications to injury mechanisms

Raúl Valldecabres[1], Ana María de Benito[2], Greg Littler[3] and Jim Richards[3]

[1] Doctorate School, Valencia Catholic University San Vicente Mártir, Valencia, Spain
[2] Physical Activity and Sports Sciences Faculty, Valencia Catholic University San Vicente Mártir, Valencia, Spain
[3] Allied Health Research Unit, University of Central Lancashire, Preston, UK

Corresponding author
Raúl Valldecabres,
raul.valldecabres@ucv.es

## ABSTRACT

The aim of this study was to determine changes in knee biomechanics during badminton lunges due to fatigue, lunge strategy and knee bracing. Kinetic and kinematic data were collected from 16 experienced right-handed badminton players. Three factor repeated measures ANOVAs (lunge direction—fatigue—brace) were performed with Least Significant Difference pairwise comparisons. In addition, clinical assessments including; $Y$-balance test, one leg hop distance and ankle dorsiflexion range of motion were performed pre- and postfatigue. The knee showed significantly greater flexion during the forehand lunge compared to backhand. In contrast, the internal rotation velocity and the knee extension moment were greater during backhand. Knee angular velocity in the sagittal plane, peak knee moment and range of moment in the coronal plane and stance time showed significantly lower values postfatigue. In addition, the peak knee adduction moment showed significantly lower values in the braced condition in both the fatigued and nonfatigues states, and no significant differences were seen for peak vertical force, loading rate, approach velocity, or in any of the clinical assessment scores. There appears to be greater risk factors when performing a backhand lunge to the net compared to a forehand lunge, and proprioceptive bracing appears to reduce the loading at the knee.

## INTRODUCTION

Badminton is characterized by high intensity effort over short durations (*Cabello, 2000*), with players needing to move quickly in multiple directions (*Jaitner & Gawin, 2007*; *Kuntze, Mansfield & Sellers, 2010*; *Sturgess & Newton, 2008*), and to execute shots while maintaining balance and motor control (*Grice, 2008*). Pivoting, jumping and lunges are the most common movements as players try to reach the shuttlecock or move back to a defensive position as quickly as possible (*Gibbs, 1988*; *Robinson & O'Donoghue, 2008*).

*Valldecabres et al. (2017)* quantified that more than 50% of lunge movements were in a diagonal direction and *Kuntze, Mansfield & Sellers (2010)* showed 15% of movements were from the center of the court to the net.

Badminton kinetics and kinematics have been previously studied (*Hong et al., 2014*; *Hong et al., 2014*; *Kuntze, Mansfield & Sellers, 2010*). However, there appears to be a lack of studies investigating the effects of fatigue, which may give a greater understanding of injury risk factors for players and coaches, and assist in the decision making during training when considering shot performance and return to sports postinjury.

During badminton 70% of injuries are to the lower limbs (*Jafari et al., 2014*; *Jørgensen & Winge, 1987*; *Shah, Ansari & Qambrani, 2014*), with approximately 50% of these being patellar tendinopathy and patellofemoral joint syndrome (*Shariff, George & Ramlan, 2009*). Extrinsic mechanisms such as; overtraining, muscle imbalance, lower extremity malalignment or knee joint laxity and training errors have all been reported as contributing factors in Patellofemoral pain (PFP) (*Tumia & Maffulli, 2002*). In addition, knee abduction moments have also been shown to be important contributors to symptoms (*Myer et al., 2015*).

Patellofemoral pain is often treated using exercise, foot orthoses, taping and knee braces (*Bolgla & Boling, 2011*). Knee braces aim to improve the tracking of the patella in the trochlea grove (*Paluska & McKeag, 2000*). The use of proprioceptive bracing in injury prevention has also attracted some attention during daily activities (*Selfe et al., 2011*) and sports specific tasks (*Hanzlíková et al., 2016*; *Sinclair et al., 2016*; *Sinclair, Vincent & Richards, 2017*); however, little is known about their efficacy when the athlete is in a fatigued state. The aim of this study was to determine the changes in knee kinetics and kinematics during badminton lunges to the net due to; fatigue, lunge direction (forehand and backhand) and knee bracing. It was hypothesized that fatigue would increase knee moments and decrease the stability during the clinical tests, whereas, knee bracing would reduce knee moments and increase the stability during the clinical tests, and that the backhand lunge would show the greatest knee moments and angular velocity. In addition, the effect of fatigue and bracing on clinical scores during dynamic stability and weight bearing tests were explored. It was hypothesized that dynamic stability during the clinical tests would decrease and angular velocity would increase during the lunge tasks following fatigue.

## MATERIALS AND METHODS

### Participants

A total of 16 right-handed badminton players (10 males and six females) with a mean age of 27.1 ± 9.0 years, height of 172.1 ± 8.9 cm and weight of 74.0 ± 16.5 kg, were recruited. All participants reported to be free from any pain or pathology affecting the lower limbs at the time of testing. This study was approved by the STEMH Ethics Committee (Ref. STEMH 671). Volunteers gave written informed consent prior to participation and all data collection conformed to the Declaration of Helsinki.

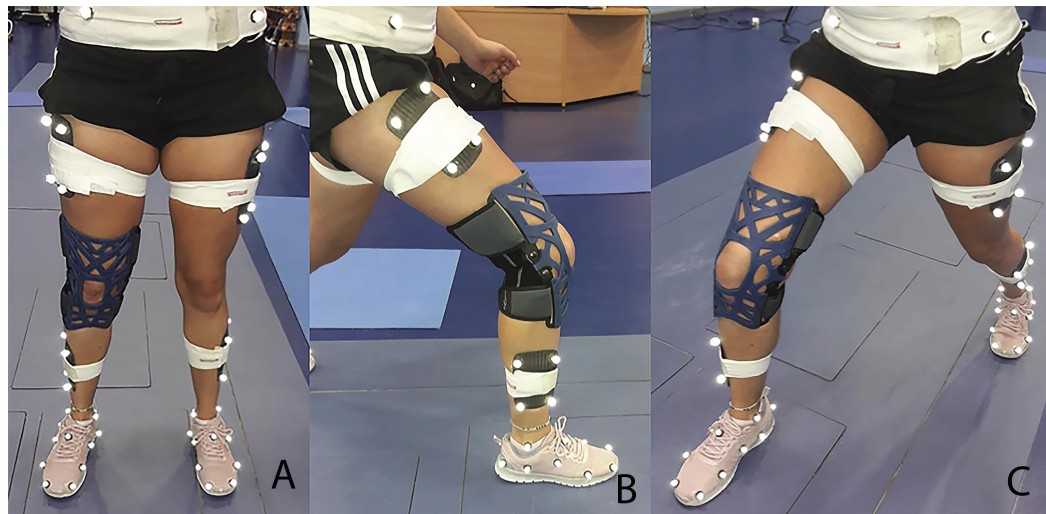

**Figure 1 Showing the lunge to the net movement with multisegment foot and marker set when wearing the knee brace.** (A) Lower limbs and multisegment foot marker set when wearing the knee brace. (B) Lunge to the net lateral view. (C) Lunge to the net medial view.

## Equipment

Kinematic data were collected using a 10 camera Oqus 7 Qualisys motion analysis system at 200 Hz (Qualisys medical AB, Gothenburg, Sweden), and kinetic data were collected at 2,000 Hz using two AMTI force platforms. Passive retroreflective markers were placed on the lower limbs using the calibrated anatomical system technique to allow for segmental kinematics to be tracked in six degrees of freedom (*Cappozzo et al., 1995*). In order to reduce measurement error, reflective markers were positioned by a single experienced researcher. Anatomical markers were positioned on the anterior superior iliac spine (ASIS), posterior superior iliac spine, greater trochanter, medial and lateral femoral epicondyle, medial and lateral malleoli and over the medial and lateral aspects of the first and fifth metatarsals. In addition, clusters of noncollinear markers were attached to the shank and thigh (Fig. 1). Markers were also placed over the forefoot, midfoot and rearfoot aspects of the shoes (Fig. 1) (*Richards, 2018*). To enable the fitting of the brace, the thigh and shank marker clusters were placed above and below the brace, respectively, as described by *Hanzlíková et al. (2016)*. Raw kinematic and kinetic data were exported to Visual3D (C-Motion Inc., Rockville, MD, USA). Kinematic and kinetic data were filtered using fourth order Butterworth filters with cut off frequencies of 15 and 25 Hz, respectively (*Hanzlíková et al., 2016*).

## Procedure

Participants were required to visit the laboratory on two occasions using a randomized order for the knee braced and no braced conditions. The knee brace used was an off the shelf proprioceptive brace (Reaction Brace; DJO Global Inc., Vista, California, USA) which was applied in accordance with the manufacturer's instructions (Fig. 1). On arrival, anthropometric measurements were taken. A standardized 10 min warm-up was performed, which included active stretching of the quadriceps and hamstring muscles

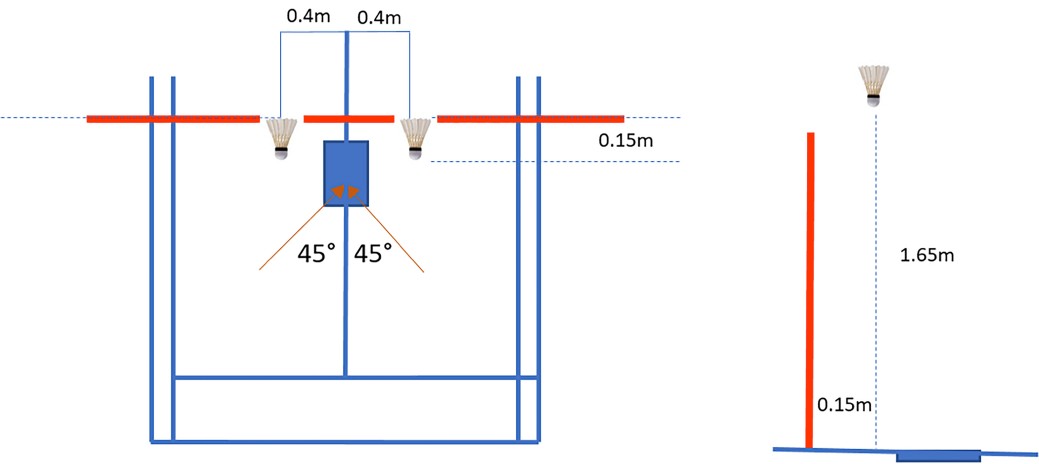

**Figure 2 Positioning of the shuttlecock with respect to the net and force plate.**

(*Lam et al., 2017*), specifically this involved five repetitions of 30 s per muscle; and familiarization of the lunge tasks, which involved performing as many repetitions as the participants needed to feel comfortable with the task (*Gribble, Hertel & Plisky, 2012*). After the warm up five lunges to the net were performed to each side (forehand and backhand), from an identical position 45° to the net. Participants were asked to hit the shuttlecock with a top spin shot, with the final step being made with the dominant limb landing on the force plate. The shuttlecock was positioned 0.15 m in front of the net, 0.4 m to the side of the force plate at a height of 1.65 m (Fig. 2). After the initial assessment, a fatigue protocol was performed which consisted of repeated forward lunges until the point of maximum volitional fatigue (*Pincivero et al., 2000*). This consisted of the lunge distance for each participant being determined as a proportion of the participants' leg length measured from the ASIS to the medial malleolus. A metronome was then used to control the number of lunges which was set to 30 repetitions per minute, a fatigued state was considered to have been reached when the participant could no longer keep up the rhythm (*Pincivero et al., 2000*). Immediately following the fatigue protocol, participants performed the lunge tasks again, the order of which was randomized. All participants wore their own sport footwear during the lunge tasks. In addition, clinical assessment tests including; the *Y*-balance test, one leg hop distance and ankle dorsiflexion range of motion test (*Weir & Chockalingam, 2007*) measured using the leg motion system (*Calatayud et al., 2015*) were conducted pre- and postfatigue state (Fig. 3).

## Data analysis

The peak vertical force, loading rate, approach velocity, stance time, and maximum, minimum, and range of motion of the knee joint angles and moments in the sagittal, coronal and transverse plane were exported from Visual3D.

## Statistical analysis

All data were examined for normality using the Shapiro–Wilks test and found suitable for parametric testing. Three factor repeated measures ANOVA tests (fatigue—lunge

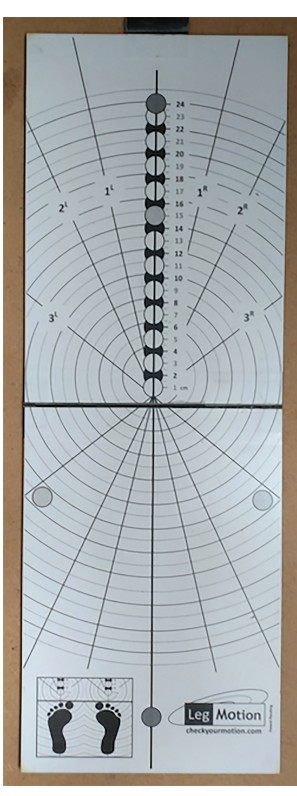

**Figure 3  Leg motion system.**

direction—brace) were performed with post hoc comparisons for the lunge tests, and two factor repeated measures ANOVA tests (fatigue—brace) were performed for the dynamic stability and weight bearing tests. In addition, the effect size was reported using Partial eta squared ($\eta_p^2$) and statistical significance was set at $p < 0.05$. All statistical analysis was performed using SPSS (v24).

## RESULTS

No significant interactions were seen between factors for any of the variables analysed. Significant main effects between pre- and postfatigue were seen in the knee flexion angular velocity at heel strike and range of knee angular velocity in the coronal plane during the lunge tasks (Table 1), with both parameters showing a 28.2 and 10.8% decrease postfatigue, respectively. In addition, significant main effects were seen in stance time, knee abduction moment and range of moment in the coronal plane (Table 2), showing 5.3, 20.2 and 8.5% lower values postfatigue, respectively (Table 3). When comparing the forehand and backhand tasks significant main effects were seen in the knee flexion angle and transverse plane knee angular velocity at heel strike (Table 1). This showed a 4.4% greater knee flexion and 66.2% lower internal rotation velocity during the forehand lunge (Table 3). In addition, significant main effects were seen in the knee extension moment (Table 2), with the forehand lunge showing a 9.0% lower knee extension moment (Table 3). When comparing the braced and no braced conditions, significant main
**Table 1 Descriptive Statistics and main effects for Knee Joint Angle and Angular Velocity.**

| Knee joint angle (degrees) | Mean ± SD | | | | | | | |
|---|---|---|---|---|---|---|---|---|
| | Pre | | | | Post | | | |
| | No brace | | Brace | | No brace | | Brace | |
| | BH | FH | BH | FH | BH | FH | BH | FH |
| Flexion at heel strike[‡] | 16.7 ± 5.4 | 18.0 ± 5.4 | 18.5 ± 5.9 | 19.7 ± 5.6 | 15.8 ± 7.5 | 17.6 ± 6.9 | 17.9 ± 5.8 | 18.1 ± 4.1 |
| Peak flexion | 64.3 ± 5.0 | 65.9 ± 7.2 | 66.8 ± 7.7 | 68.7 ± 8.8 | 62.9 ± 7.2 | 64.4 ± 8.4 | 65.4 ± 6.6 | 66.2 ± 7.8 |
| ROM sagittal plane | 47.6 ± 4.6 | 47.7 ± 5.6 | 48.3 ± 6.3 | 48.9 ± 7.0 | 47.2 ± 5.6 | 46.8 ± 5.4 | 47.4 ± 5.6 | 48.0 ± 5.7 |
| Coronal plane at heel strike | −0.7 ± 3.8 | 0.2 ± 4.1 | 0.5 ± 5.3 | 0.4 ± 5.1 | −0.4 ± 3.8 | −0.4 ± 4.0 | 0.3 ± 5.0 | −0.2 ± 4.6 |
| Peak valgus | 3.6 ± 6.0 | 4.4 ± 6.2 | 4.0 ± 8.0 | 4.4 ± 7.6 | 3.7 ± 5.8 | 3.5 ± 5.5 | 3.8 ± 7.8 | 3.2 ± 7.8 |
| Peak varus | −4.1 ± 5.8 | −3.4 ± 5.4 | −4.6 ± 6.1 | −4.6 ± 5.7 | −4.0 ± 4.9 | −4.2 ± 4.8 | −5.4 ± 6.2 | −5.4 ± 5.8 |
| ROM coronal plane | 7.6 ± 3.7 | 7.7 ± 3.2 | 8.6 ± 5.3 | 9.0 ± 5.2 | 7.7 ± 3.1 | 7.7 ± 3.2 | 9.2 ± 5.3 | 8.6 ± 5.2 |
| Transverse plane at heel strike | −2.9 ± 9.1 | −5.5 ± 8.5 | −5.1 ± 10.9 | −5.0 ± 11.5 | −4.9 ± 8.5 | −3.8 ± 8.0 | −5.7 ± 10.6 | −5.3 ± 10.6 |
| Peak external rotation | 7.6 ± 6.0 | 7.7 ± 4.6 | 6.2 ± 7.9 | 6.4 ± 8.4 | 6.9 ± 4.6 | 7.6 ± 4.5 | 6.1 ± 7.4 | 6.3 ± 7.3 |
| Peak internal rotation | −5.6 ± 8.1 | −7.2 ± 7.4 | −7.3 ± 10.5 | −6.8 ± 11.0 | −7.7 ± 7.5 | −6.4 ± 7.1 | −7.1 ± 10.5 | −6.8 ± 10.1 |
| ROM transverse plane | 13.2 ± 6.2 | 14.9 ± 5.8 | 13.5 ± 6.0 | 13.2 ± 4.4 | 14.6 ± 6.3 | 14.0 ± 5.3 | 13.3 ± 6.8 | 13.1 ± 5.4 |
| **Knee joint angular velocity (degrees·seconds$^{-1}$)** | | | | | | | | |
| Flexion velocity at heel strike* | 175.7 ± 114.5 | 197.6 ± 137.6 | 195.2 ± 110.6 | 200.2 ± 76.0 | 121.0 ± 152.8 | 159.9 ± 133.0 | 115.9 ± 85.5 | 155.3 ± 107.0 |
| Peak flexion angular velocity | 535.3 ± 73.2 | 524.4 ± 91.0 | 515.3 ± 90.3 | 509.1 ± 83.6 | 546.8 ± 84.9 | 531.1 ± 68.5 | 532.2 ± 78.3 | 522.0 ± 84.4 |
| Range of velocity sagittal plane | 539.8 ± 84.1 | 532.0 ± 99.4 | 517.3 ± 95.2 | 508.9 ± 81.5 | 571.4 ± 130.1 | 539.0 ± 66.7 | 538.3 ± 79.9 | 522.8 ± 83.5 |
| Valgus velocity at heel strike | 4.6 ± 66.6 | −15.3 ± 70.3 | −38.2 ± 79.6 | −43.2 ± 89.3 | 0.9 ± 54.5 | −2.2 ± 48.2 | −29.5 ± 93.8 | −22.4 ± 64.0 |
| Peak valgus velocity | 191.4 ± 117.9 | 229.4 ± 106.2 | 206.4 ± 137.0 | 215.8 ± 136.7 | 192.7 ± 114.1 | 202.7 ± 100.8 | 188.4 ± 131.5 | 196.0 ± 127.9 |
| Peak varus velocity | −96.5 ± 55.0 | −115.5 ± 45.5 | −131.5 ± 69.9 | −140.7 ± 55.8 | −87.7 ± 39.6 | −103.8 ± 41.9 | −109.8 ± 69.3 | −102.8 ± 45.0 |
| Range of velocity coronal plane* | 287.9 ± 136.8 | 344.9 ± 125.0 | 337.9 ± 159.5 | 356.5 ± 142.3 | 280.4 ± 120.9 | 306.5 ± 111.7 | 298.2 ± 150.6 | 298.7 ± 138.1 |
| Transverse plane velocity at heel strike[‡] | 11.0 ± 125.2 | −50.9 ± 106.4 | 63.3 ± 205.8 | 30.3 ± 136.9 | 9.5 ± 157.3 | −28.8 ± 135.4 | 28.6 ± 139.7 | 11.4 ± 143.3 |
| Peak internal rotation velocity | 298.0 ± 114.4 | 345.7 ± 122.8 | 346.3 ± 178.7 | 329.8 ± 127.9 | 312.0 ± 128.8 | 320.7 ± 150.6 | 347.0 ± 178.7 | 329.8 ± 127.9 |
| Peak external rotation velocity | −166.7 ± 81.8 | −149.1 ± 62.3 | −164.4 ± 73.6 | −165.7 ± 69.1 | −162.1 ± 96.0 | −151.0 ± 79.6 | −132.0 ± 80.4 | −122.0 ± 49.8 |
| Range of velocity transverse plane | 464.7 ± 147.8 | 494.8 ± 146.8 | 510.7 ± 182.1 | 495.5 ± 155.2 | 474.1 ± 149.7 | 471.7 ± 188.4 | 479.0 ± 205.4 | 447.6 ± 141.4 |

Notes:
* Significant main effect between pre- and postfatigue.
‡ Significant main effect between Backhand (BH) and Forehand (FH).

effects were seen in the peak knee adduction moment (Table 2), with a 34.8% lower knee moment being seen in the braced condition (Table 3). For the force and time data no significant effects were seen for peak vertical force, loading rate or approach velocity.

**Table 2 Descriptive Statistics and main effects for knee moments.**

| Knee moments (N/m) | Mean ± SD | | | | | | | |
| --- | --- | --- | --- | --- | --- | --- | --- | --- |
| | Pre | | | | Post | | | |
| | No brace | | Brace | | No brace | | Brace | |
| | BH | FH | BH | FH | BH | FH | BH | FH |
| Peak flexion moment | 138.4 ± 56.0 | 149.1 ± 58.3 | 141.2 ± 49.9 | 139.6 ± 51.9 | 136.0 ± 52.1 | 141.3 ± 56.2 | 134.8 ± 40.8 | 138.6 ± 45.9 |
| Peak extension moment[‡] | −55.0 ± 24.2 | −45.0 ± 20.7 | −51.9 ± 31.8 | −50.7 ± 35.3 | −54.5 ± 19.7 | −49.6 ± 23.5 | −60.1 ± 26.3 | −56.1 ± 25.2 |
| Sagittal plane moment range | 193.4 ± 68.6 | 194.1 ± 66.0 | 193.1 ± 71.1 | 190.4 ± 75.2 | 190.5 ± 59.1 | 190.9 ± 68.0 | 195.0 ± 58.5 | 194.6 ± 62.7 |
| Peak abduction moment | 75.3 ± 58.1 | 74.6 ± 64.9 | 70.4 ± 43.8 | 67.5 ± 43.8 | 62.1 ± 35.9 | 76.5 ± 69.6 | 66.1 ± 38.6 | 68.6 ± 38.4 |
| Peak adduction moment[*†] | −22.8 ± 33.7 | −25.8 ± 33.4 | −16.2 ± 37.1 | −19.9 ± 45.6 | −21.5 ± 32.4 | −21.9 ± 34.1 | −11.6 ± 41.3 | −12.5 ± 44.5 |
| Coronal plane moment range[*] | 98.1 ± 50.3 | 100.3 ± 58.1 | 86.5 ± 51.0 | 87.4 ± 52.4 | 83.6 ± 33.3 | 98.4 ± 62.7 | 77.7 ± 42.1 | 81.1 ± 46.1 |
| Peak internal rotation moment | 11.9 ± 9.2 | 13.6 ± 16.5 | 9.4 ± 4.3 | 12.7 ± 8.3 | 11.5 ± 6.6 | 13.3 ± 16.3 | 10.7 ± 5.1 | 13.2 ± 8.7 |
| Peak external rotation moment | −3.5 ± 3.4 | −3.7 ± 3.42 | −4.6 ± 3.3 | −3.5 ± 4.7 | −3.2 ± 3.5 | −3.6 ± 3.4 | −4.0 ± 7.2 | −2.3 ± 4.8 |
| Transverse plane moment range | 15.4 ± 8.2 | 17.3 ± 18.0 | 14.1 ± 4.2 | 16.1 ± 7.2 | 14.8 ± 5.6 | 16.9 ± 17.2 | 14.7 ± 6.8 | 15.5 ± 7.7 |
| **Force and time** | | | | | | | | |
| Peak vertical force (N) | 1293.8 ± 448.2 | 1305.6 ± 424.3 | 1293.2 ± 506.1 | 1184.1 ± 515.3 | 1274.0 ± 421.2 | 1305.9 ± 508.1 | 1255.2 ± 482.9 | 1171.8 ± 504.4 |
| Loading rate (BW/s) | 575.2 ± 214.0 | 525.8 ± 171.7 | 625.6 ± 304.5 | 557.0 ± 232.1 | 611.6 ± 203.1 | 603.3 ± 163.4 | 571.8 ± 293.2 | 521.3 ± 271.1 |
| Approach velocity (m/s) | 2.2 ± 0.3 | 2.1 ± 0.4 | 2.2 ± 0.3 | 2.2 ± 0.2 | 2.1 ± 0.3 | 2.0 ± 0.5 | 2.0 ± 0.4 | 2.1 ± 0.3 |
| Right stance time (s)[*] | 0.242 ± 0.33 | 0.242 ± 0.42 | 0.257 ± 0.06 | 0.246 ± 0.07 | 0.225 ± 0.04 | 0.229 ± 0.05 | 0.249 ± 0.07 | 0.231 ± 0.53 |

Notes:
[*] Significant main effect between pre- and postfatigue.
[†] Significant main effect between brace and no brace.
[‡] Significant main effect between Backhand (BH) and Forehand (FH).

No significant differences were seen between pre- and postfatigue or between brace and no brace for the *Y*-balance test, one leg hop distance or ankle dorsiflexion range of motion test (Table 4).

**Table 3 Post hoc analysis for Significant Main effects for joint angle, angular velocity and knee moments.**

| Pre vs. post | Pre vs. post | | p-value | Confidence intervals of the difference | Effect size $\eta_p^2$ |
|---|---|---|---|---|---|
| | Mean difference | | | | |
| | Pre | Post | | | |
| Flexion at heel strike | 192.2 | 138.0 | 0.003 | 21.963–86.331 | 0.460 |
| Range of velocity coronal plane | 331.8 | 296.0 | 0.011 | 9.565–62.138 | 0.360 |
| Peak varus moment | −21.2 | −16.9 | 0.004 | −6.989 to −1.599 | 0.435 |
| Coronal plane moment range | 93.1 | 85.2 | 0.034 | 0.673–15.093 | 0.266 |
| Right stance time | 0.247 | 0.234 | 0.012 | 0.003–0.023 | 0.352 |
| **BH vs. FH** | **BH vs. FH** | | **p-value** | **Confidence intervals of the difference** | **Effect size $\eta_p^2$** |
| | Mean difference | | | | |
| | BH | FH | | | |
| Flexion at heel strike | 17.4 | 18.2 | 0.047 | −2.274 to −0.015 | 0.240 |
| Transverse plane velocity at heel strike | 28.1 | −9.5 | 0.012 | 9.500–65.806 | 0.350 |
| Peak extension moment | −55.4 | −50.4 | 0.019 | −9.110 to −0.966 | 0.317 |
| **No brace vs. brace** | **No brace vs. brace** | | **p-value** | **Confidence intervals of the difference** | **Effect size $\eta_p^2$** |
| | Mean difference | | | | |
| | No brace | Brace | | | |
| Peak adduction moment | −23.0 | −15.0 | 0.028 | −14.953 to −0.993 | 0.283 |

**Table 4 Dynamic stability test and weight bearing results for right stance leg.**

| | Brace | | No brace | | p-value | Effect Size $\eta_p^2$ | p-value | Effect size $\eta_p^2$ |
|---|---|---|---|---|---|---|---|---|
| | Mean ± SD | | Mean ± SD | | Pre–Post | | Brace–No Brace | |
| | Pre | Post | Pre | Post | | | | |
| Weight bearing (cm) | 10.8 ± 3.4 | 10.4 ± 3.3 | 10.1 ± 2.9 | 10.3 ± 3.2 | 0.721 | 0.009 | 0.200 | 0.107 |
| Y-test anterior (m) | 0.7 ± 0.1 | 0.7 ± 0.1 | 0.7 ± 0.8 | 0.7 ± 0.1 | 0.266 | 0.082 | 0.427 | 0.043 |
| Y-test Posteromedial (m) | 1.1 ± 0.2 | 1.1 ± 0.2 | 1.1 ± 0.1 | 1.0 ± 0.1 | 0.873 | 0.002 | 0.465 | 0.036 |
| Y-test Posterolateral (m) | 0.9 ± 0.2 | 1.0 ± 0.2 | 0.9 ± 0.2 | 1.0 ± 0.2 | 0.080 | 0.190 | 0.593 | 0.02 |
| One leg hop distance (cm) | 115.9 ± 23.1 | 116.6 ± 23.0 | 121.2 ± 20.9 | 115.4 ± 22.0 | 0.373 | 0.053 | 0.530 | 0.027 |

## DISCUSSION

The aim of the current investigation was to examine the effects of fatigue, lunge strategy and wearing a knee brace on knee kinetics and kinematics during badminton lunges to the net and clinical scores in experienced badminton players. Key findings for the effect of fatigue showed that the knee flexion angular velocity at heel strike, range of knee angular velocity in the coronal plane decreased in a fatigue state. Kinetic data showed that the peak knee adduction moment and coronal plane moment range were all lower postfatigue, which occurred over a shorter stance time. The changes in joint angular velocity, with no

corresponding change in joint angles, would indicate that there is a slower movement, however, no significant difference was seen in the approach speed. Therefore, this would indicate an increase in joint stiffness in the sagittal and coronal planes defined by *Hughes & Watkins (2008)* with a lower adaptability as the leg resistance moves into compression over less time during landing. This increase in stiffness is supported by *Arampatzis et al. (2001)* who found that lower limbs stiffness influences athletic performance in sports activities. This could relate to a potential increase in injury risk due to increase stress and strain in the knee joint (*Derrick, Dereu & Mclean, 2002*; *Dierks, Davis & Hamill, 2010*) and changes to dynamic loads on the lower limbs through an interaction of simultaneous concentric and eccentric contractions when athletes are in a fatigue state (*Komi, 2000*). One explanation for the decreases in peak knee adduction moment and coronal plane moment range, could be a change in strategy during loading, which may relate to changes in foot position and posture during the lunge. This reduction in the knee adduction moments could be explained by the foot landing in more external rotated position, therefore changing the line of action of the ground reaction force; although no changes were seen in the transverse plane moments at the knee. However, further exploration of such compensatory mechanisms due to foot placement is beyond the scope of this current paper.

When comparing the forehand and backhand tasks significant main effects were seen in the sagittal and transverse planes. During the forehand lunge a greater knee flexion was seen at heel strike with less internal rotation than the backhand lunge. This would indicate a lower injury risk during the forehand lunge, as increases in internal rotation movements have been shown to be an anterior cruciate ligament injury risk mechanism (*Fornalski et al., 2008*; *Myer et al., 2008*; *Meyer & Haut, 2008*).

When comparing the braced and no braced conditions, a significant reduction in peak knee adduction moment was seen in the braced condition (Tables 2 and 3). This would indicate a reduction in the medial compartment contact force (*Manal et al., 2015*), which has been associated with lower pain levels in knee OA and reductions in knee varum (*Miyazaki, 2002*). However, the brace used in this study was not a rigid brace and therefore this effect is unlikely to be from any mechanical realignment of the knee, but can be explained by a change in loading strategy due to changes in proprioception. This has been previously seen in several studies during step descent (*Akseki, 2008*; *Baker et al., 2002*; *Callaghan et al., 2002*, *2008*; *Selfe et al., 2011*), and sports related movement tasks (*Hanzlíková et al., 2016*; *Sinclair et al., 2016*), who reported improvements in knee stability and reductions in knee pain.

Interestingly no significant differences were seen between pre- and postfatigue or between brace and no brace for the Y-balance test, one leg hop distance or ankle dorsiflexion range of motion test. This would indicate that overall performance was unchanged, whereas movement control and strategy during the lunge tasks were affected. This suggests that these clinical scores were not sensitive to potentially clinically important changes that can be associated with knee injury risk factors.

Limitations of this study include; participants wearing their own shoes rather than standardized footwear. Although *Park et al. (2017)* suggested that different designs of

badminton shoes do not significantly affect lower extremity kinematics, although these did have an effect on subjective perception of comfort. In addition, this study recruited participants who were recreational athletes who had played badminton for at least 2 years, however, due to possible differences in technique it is not possible to extrapolate these findings to elite players.

## CONCLUSIONS

This study showed no significant differences in approach velocity and loading rate postfatigue, however, a greater knee stiffness was seen. In addition, there appears to be greater risk factors when performing a backhand lunge to the net compared to a forehand lunge. These factors should be considered when developing training regimes.
Finally, proprioceptive bracing appears to improve the loading patterns at the knee, which should be considered when players are returning to sport after an injury.

## ACKNOWLEDGEMENTS

We appreciate the help of Laurence Smith and Komsak Sinsurin when collecting data during this study.

### Funding

This work was supported by Generalitat Valenciana ACIF projects [Grants number ACIF2016/121 and BEFPI 2017/014] and Valencia Catholic University 'San Vicente Mártir' pre-competitive grants for research groups. The funders had no role in study design, data collection and analysis, decision to publish, or preparation of the manuscript.

### Grant Disclosures

The following grant information was disclosed by the authors:
Generalitat Valenciana ACIF projects: ACIF2016/121 and BEFPI 2017/014.
Valencia Catholic University San Vicente Mártir.

### Competing Interests

The authors declare that they have no competing interests.

### Author Contributions

- Raúl Valldecabres conceived and designed the experiments, performed the experiments, analyzed the data, contributed reagents/materials/analysis tools, approved the final draft.
- Ana María de Benito performed the experiments, contributed reagents/materials/analysis tools, approved the final draft.
- Greg Littler prepared figures and/or tables, authored or reviewed drafts of the paper, approved the final draft.
- Jim Richards conceived and designed the experiments, performed the experiments, analyzed the data, contributed reagents/materials/analysis tools, approved the final draft.

## Human Ethics

The following information was supplied relating to ethical approvals (i.e., approving body and any reference numbers):

STEMH Ethics Committee Application (Unique Reference Number: STEMH 671).

## Data Availability

The raw data are provided in the Supplemental Files.

## Supplemental Information

Supplemental information for this article can be found online at http://dx.doi.org/10.7717/peerj.6033#supplemental-information.

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
