# Peer review of "An exploration of the effect of proprioceptive knee bracing on biomechanics during a badminton lunge to the net, and the implications to injury mechanisms"

_PeerJ, doi:10.7717/peerj.6033_

## Round 0.1 · original submission · Major Revisions

The three reviewers and I see much merit in this paper but have highlighted some areas that need to be improved for it to be considered for publication in PeerJ.

Reviewer 1 ·

Basic reporting

.

Experimental design

.

Validity of the findings

.

Additional comments

LINE 48: When you first write abbreviations, you must describe it.
LINE 72-74: It would have been better to attach the photo the article” Reflective markers were placed on the shank and thigh using the Calibrated Anatomical System Technique” and” To enable the fitting of the brace, the thigh and shank marker clusters were placed above and below the brace respectively”
LINE 78: Please upload the photo ”knee braced”
LINE 81: How is “standardised warm-up”?
LINE 83: It would have been better to have clear pictures of how to do the test.
LINE159: Please list numerous studies for ”This has been previously seen in several studies during step descent (Selfe et al., 2011)”
Table1: What does HS mean? Please write the full term.
It is better to refer to the results of Table 2 and 3 for more positive brace effect
The English language should be improved.

Reviewer 2 ·

Basic reporting

Meet journal's standards.

Experimental design

Experimental design is acceptable, with sufficient detail to replicate.

Validity of the findings

Data is robust, statistically sound, & controlled.

Additional comments

1. In the introduction part, second paragraph, line 42-43, “This may provide important information for players and coaches to assist in the decision making in training when considering shot performance and return to sports post injury.” What does “this” refer to? I suggest to combine this sentence with the previous one, for instance, “…which may give a greater understanding of injury risk factors and provide important information for players and coaches to assist in the decision making in training when considering shot performance and return to sports post injury.”

2. In the introduction part, third paragraph, line 48, what does “PFP” refer to? If this is the first abbreviation, please mark the full name before giving the abbreviation.

3. In the introduction part, third paragraph, line 49, “knee abduction moments have also been shown to be an important contributor to symptoms.” Grammar mistakes exist in this sentence, which should be replaced by “knee abduction moment has also been shown to be an important contributor to symptoms.” Please modify.

4. In the discussion part, third paragraph, line 154-155, “This would indicate a reduction the medial compartment contact force.” grammar mistakes exist in this sentence, which should be replaced by “This would indicate a reduction in the medial compartment contact force.” Please modify.

5. In the discussion part, fourth paragraph, line 164-165, “…whereas movement control and strategy during the lunge tasks was affected.” Grammar mistakes exist in this sentence, which should be replaced by “…whereas movement control and strategy during the lunge tasks were affected.” Please modify.

6. In the discussion part, last paragraph, line 171-172, “it is not possible the extrapolate these findings to elite players.” Grammar mistakes exist in this sentence, which should be replaced by “it is not possible to extrapolate these

findings to elite players.” Please modify.

7. Grammar mistakes were not mentioned completely, please pay more attention and check it carefully throughout this manuscript.

8. Please pay more attention on the format of this manuscript and make revision based on the journal guidelines.

Reviewer 3 ·

Basic reporting

manuscript is reasonably well written - language. but some parts are missing

Experimental design

is ok, power analysis - sample size estimation needed and reliability of experimental protocol and analysis is required.

Validity of the findings

I have some concerns - see specific comments

Additional comments

General comments
In this manuscript, a study investigating the effects of a knee brace and fatigue on the biomechanics of a forward lunge to the net in badminton is presented. The study design is rather complex as three factors (with and without brace, pre and post fatigue and backhand and forehand lunge) were to be investigated by testing 16 subjects repeatedly while fatiguing them during each of two sessions. The paper as such is well written and the study itself is relevant and might add to the knowledge in regard to fatigue effects and knee injury potential in badminton. There are, however, major concerns in regard to the methods section which appears incomplete. In particular, there is no mention, reference or discussion on the reliability of the marker placement protocol which are crucial factors. I also feel that the results section is quite heavy and not easy to digest. Too many mean values are presented in the tables. While not being wrong as such it becomes very tedious to read. I would also prefer to see some moment curves instead, to highlight the time curves where significant differences occur. Rather stick to a presentation with main and significant findings (as indicated above – in relation to specific hypotheses), like in Tables 3 and 4. The discussion appears to contain some misconceptions and remains on a very general level. This needs to be substantially improved.

Specific comments
Abstract (first time listed in the review document):
Line 5 specify the type of post hoc test used.
Line 10 … showed significantly lower …
Third last From the wording it is not immediately clear if the clinical assessments were done at pre and post or only once each session, reword!
Second last what it meant by an ‘improved loading pattern’ needs to be defined.

Introduction:
Line 48 Acronym PFP needs to be introduced.
Line 56 Aim is not quite clear: you now list fatigue, lunge strategy and knee bracing as factors. It should read ‘lunge direction, forehand and backhand’. Further, you list a number of parameters to be looked at while some clear hypotheses would make it easier for the reader but also yourselves how to present results and so on. Provide specific hypotheses which can be tested statistically.
Methods:
Line 72 Spelling of several authors is erroneous, correct here and in the reference list.
Line 74 More information is needed how the data were further processed in visual 3D, e.g., which model was used or did you use a custom model? How did you extract peak values etc.? How reliable is your experiment – the calculations? This is needed to assess if the differences are, e.g., clinically relevant.
Line 75 Why these and why different cut-off frequencies. Provide a reason or explain how you estimated these?
Line 78 … the laboratory on two occasions …
Line 89 Following the fatigue protocol, participants …
Line 90/91 Neither here nor in the abstract it is (grammatically) 100% clear if the additional tests were only carried out once per session or pre and post fatigue. If the latter (which your results indicate), it needs to be indicated how long it took from end of fatiguing protocol to the tests, also how long until the 5 lunges were performed. A definition of fatigue is missing. Did you test for fatigue, e.g., force measured, perception tests? Also many more details are needed. Did the markers stay on the subjects for the whole session? Did you control if the moved or fell off (not sure if they did but it may happen due to sweating - please comment)?
Line 98 Referring to a comment above: I presume that SPSS was only used to statistically compare results (hence the title of this paragraph should rather read 'statistical analysis', while the data analysis in visual 3D should be more detailed. 
Results:
Line 108 you only report main effects but shouldn’t you first report interactions? Or were none of them significant?
More general I don’t think it is pleasant to be simply referred to the tables. I gather it would be clearer to concentrate the tables to the hypothesis-derived numbers or merely list significant findings. You can then indicate in the text %-differences or F statistics as recommended in many other journals. – The results appear comprehensive but as if you compared almost everything possible and ran statistics to check where you might find differences. Apart from the fact that you inflate the number of comparisons and with that the chance for type 1 errors you should also present a sample size estimation of sorts.
Discussion:
Line 127 … net moments … ??
Line 143, end ‘… plane’ or better ‘… in the sagittal and transverse planes’.
Line 146 This thought appears unrelated. I don't think that a lunge would lead to hyperextension of the knee, also you mention that there is an increase knee flexion, that would be a contradiction in itself. So how is this possible? 
Try to rework the discussion much more specifically. What differences did you identify and what do they mean in regard to joint loading etc.?
Line 149 … five times higher risk for ACL injury.
Line 151 There seems to be a misconception. An extension moment does not imply a hyperextension of the knee.
Line 154 … reduction of the medial compartment …
Line 157 … mechanical realignment …
Line 171, end … not possible to extrapolate …
Line 177 Finally, proprioceptive …
Line 178 This is a somewhat long shot: You investigated the effect of braces, how does this play into how to develop training strategies?

---

## Round 0.2 · Minor Revisions

General Comments

Thanks for addressing most of the concerns of the reviewers. The following small issues still need to be addressed. All of my comments reflect the line numbers of the track changes document.

Specific Comments

line 72 – 77: I don’t think you have yet clearly presented the hypotheses as yet. Please make it more explicit how the knee kinetics and kinematics may change as a function of the three factors.
Line 103: please remove “Hanzlíková et al” from within the parentheses and place it before the year of publication.
line 113 – 117: it still is not completely clear what was involved in the warmup; as I don’t think I could replicate this warmup based on what you have described so far. Please provide more detail about the types of stretches (passive or active), durations of stretches and what actually happened in the familiarisation with the lunges with respect to sets, reps, instructions etc.
Line 144: I suggest you break this into two parts, being the current Data Analysis and a new Statistical Anlaysis section. The Statistical Analysis should start the sentence on line 140 – 141 regarding the Shapiro Wilks and then have the three factor repeated measures ANOVA, followed by mention of the p values and SPSS.
Line 199 – 201: I suggest you remove this whole sentence about the hyperextension as you did not see any such results in your study.
Overall Discussion: consistent with some of the reviewers, I still feel that the discussion is a bit limited in its scope and depth of analysis of your results compared to the literature. I suggest you add some additional detail regarding your kinetic data and what it suggests regarding the knee joint loading across the three independent variable assessments within this study.

Reviewer 2 ·

Basic reporting

Clear and unambiguous, professional English used throughout.

Experimental design

Original primary research within Aims and Scope of the journal.

Validity of the findings

Conclusion are well stated, linked to original research question & limited to supporting results.

Additional comments

Revised version is suitable to publish.

---

## Round 0.3 · accepted · Accept

I thank the authors for attending to the final constructive criticisms of the previous manuscript and am happy to inform you that I am recommending it be accepted for publication in PeerJ.